# Large language models for thematic analysis in healthcare research: A blinded mixed-methods comparison with human analysts

Callum Hill ⬤*, Arun Dahil ⬤, Glenn Simpson, David Hardisty, Jacob Keast, Cameron Kumar Pinn ⬤, Hajira Dambha-Miller

Primary Care Research Centre, University of Southampton, Southampton, United Kingdom

* ch1g22@soton.ac.uk

## Abstract

Large language models (LLMs) are increasingly used for qualitative thematic analysis, yet evidence on their performance in analysing focus-group data, where polyvocality and context complicate coding, remains limited. Given the increasing role of such models in thematic analysis, there is a need for methodological frameworks that enable systematic, metric-based comparisons between human and model-based analyses. We conducted a blinded mixed-methods comparison of two general-purpose LLMs (ChatGPT-5 and Claude 4 Sonnet), an LLM-based qualitative coding application (QualiGPT), and blinded human analysts on an in-person focus-group transcript informing an AI-enabled digital health proposal. We evaluated deductive coding using a 10-code, 6-theme codebook against an expert consensus adjudication; inductive coding with a structured Likert-scale comparison to a reference-standard set of inductive themes generated by expert consensus; and manual quote verification of LLM segments to define LLM hallucination (evidence absent or non-supportive) and error rate (including partial matches and speaker-coded segments). During deductive coding against an expert consensus adjudication, large language models yielded a mean agreement of 93.5% (95% CI 92.5–94.5) with κ = 0.34 (95% CI 0.26–0.40); blinded human coders achieved 92.7% (95% CI 91.6–93.9) agreement with κ = 0.34 (95% CI 0.26–0.41). Mean Gwet's AC1 was 0.92 (95% CI 0.90–0.93) for the blinded human analysis, and 0.93 (95% CI 0.92–0.94) for the LLM-assisted deductive analysis, reflecting high agreement despite the low overall code prevalence (7.8%, SD = 3.2%). Only one model achieved non-inferiority in inductive analysis of the transcript (p = 0.043). The strict hallucination rate in inductive analysis was 1.2% (SD = 2.1%). LLMs were non-inferior to human analysts for deductive coding of the focus-group data, with variable performance in inductive analysis. Low hallucination but significant comprehensive error rates indicate that LLMs can augment qualitative analysis but require human verification.

which permits unrestricted use, distribution, and reproduction in any medium, provided the original author and source are credited.

**Data availability statement:** The primary data that support the findings of this study are not publicly available due to privacy and ethical considerations, as despite anonymisation, the transcript contains potentially identifiable information. Full results of thematic analysis including excerpts from the transcript are available in the supporting material. Code used in data processing and analysis is publicly available on GitHub: https://github.com/CHill887/Large-Language-Models-for-Thematic-Analysis-Mixed-Methods-Comparison.

**Funding:** This report is independent research funded by the National Institute for Health Research (Artificial Intelligence for Multiple Long-Term Conditions (AIM), "The development and validation of population clusters for integrating health and social care: A mixed-methods study on Multiple Long-Term Conditions," NIHR202637). HDM receives funding from the National Institute for Health and Care Research (NIHR) Multiple Long-Term Conditions (MLTC) Cross NIHR Collaboration (CNC) (NIHR207000). The views expressed in this publication are those of the author(s) and not necessarily those of the NHS, the NIHR or the Department of Health and Social Care. The funders had no role in study design, data collection and analysis, decision to publish, or preparation of the manuscript.

**Competing interests:** The authors have declared that no competing interests exist.

**Abbreviations:** AI, Artificial intelligence; AC1, Agreement Coefficient 1 (Gwet's); CI, Confidence interval; GDPR, General Data Protection Regulation; GRAMMS, Good Reporting of a Mixed Methods Study; LLM, Large language model; NIHR, National Institute for Health and Care Research; NHS, National Health Service; PPI, Patient and Public Involvement; SD, Standard deviation.

## Author summary

Qualitative research plays an important role in digital health, assisting in the implementation of healthcare technologies and innovations. However, analysing qualitative data in the form of focus groups is time-consuming and requires human expertise. Large Language Models (LLMs) are being increasingly used in qualitative research analysis, although evidence on their performance in analysing focus group data is limited. We compared the performance of LLMs to blinded human analysts in analysing a focus group transcript on AI implementations in healthcare. We used both qualitative and quantitative metrics to evaluate the performance of LLMs in thematic analysis.

We found that the LLMs performed similarly to humans when applying pre-defined codes (deductive analysis), with a low rate of hallucination. However, in open-ended theme generation (inductive analysis) their performance was more variable, particularly in areas requiring interpretation of tone, nuance, or conversational context. These findings suggest that LLMs can be used to support interpretation of qualitative data, rather than replace human analysts. We provide a reproducible framework in analysing the performance of LLMs in qualitative analysis.

## Introduction

Due to recent developments in data computation and scaling, large language models (LLMs) have advanced significantly since their inception, now demonstrating the capacity to analyse human text and infer both explicit and implicit meaning [1,2]. Understanding whether LLM-assisted analysis can reliably surface patient and clinician-centred requirements that inform clinical decision support design is directly relevant to their deployment within qualitative clinical research settings. There is growing interest in the existing literature surrounding the utility of LLMs in thematic analysis, due to their rapid processing of data, as well as their potential to mitigate intrinsic researcher biases in qualitative research- an important consideration when exploring patient experiences, clinician perspectives, and healthcare decision-making processes [3–5].

Existing work has demonstrated the performance of LLMs in a range of qualitative analysis tasks [6,7]. Bennis et al demonstrated high researcher agreement with LLMs when analysing survey responses [8]. Kornblith et al compared LLMs against parallel human analysts in both sentiment classification and thematic categorisation, finding comparable inter-human and LLM-human agreement [9].

Previous studies have focused largely on interview data, with minimal exploration of focus group transcripts [10–12]. To our knowledge, there have been no studies comparing the use of LLMs to human researchers in thematic analysis of focus group data in a healthcare context. Focus groups are widely used in healthcare research to capture co-constructed meaning, patient-clinician interactions, and shared understanding of care processes.

Qualitative coding of focus group data presents several challenges to LLMs, as compared to interview data. Interview transcripts are typically linear, dyadic, and contextually consistent. In contrast, focus groups are polyvocal, featuring a greater prevalence of co-constructed and co-dependent meaning which relies more heavily on interpersonal and contextual meaning [12]. Prior work has suggested that LLMs are more limited in their ability to identify latent meaning, interpersonal dynamics, and contextual meaning [13]. Finally, focus groups can often generate "noise" from tangential dialogue [12].

Methodological approaches comparing human and LLM interpretations have varied widely. Studies differ in their use of common global coding rules, researcher blinding, and evaluation metrics, ranging from primarily quantitative comparisons to subjective qualitative impressions [13–16]. Input datasets in LLM analyses are often simplified for ease of analysis, meaning that results are not always representative of results in practical analytic tasks [17]. Finally, it is well documented that LLMs "hallucinate" data or text in their responses, as this property has been described as intrinsic to their design [18,19]. Few studies have quantified the effect of this property on qualitative analysis.

Collectively, prior studies suggest that LLMs tend to perform strongly in structured or deductive coding tasks where themes are explicitly defined yet demonstrate more variable performance when analysis requires interpretation of latent meaning, interpersonal dynamics, or contextual nuance. Furthermore, methodological approaches in existing studies range from subjective qualitative comparisons to purely quantitative agreement metrics, making cross-study comparisons difficult. Additionally, a significant proportion of the current literature relies on interview or survey data, which lack the polyvocal and interactional complexity characteristic of focus groups frequently used in healthcare research. These gaps highlight the need for evaluative frameworks that combine quantitative agreement metrics with qualitative interpretive comparison in realistic analytic settings, enabling a clearer assessment of the utility of LLM-assisted thematic analysis in healthcare research.

This study aims to assess the performance of LLMs using an existing database of qualitative focus group and patient and public involvement (PPI) sessions focused on the development of an AI tool to support physical activity in the management of chronic conditions. To our knowledge, this study represents the first mixed-methods study in a healthcare focus group context, comparing LLMs and humans in thematic analysis with formalised evaluations of inductive, deductive, and error-based performance.

## Materials and methods

### Ethics statement

Ethical approval was granted by Southampton University Faculty of Medicine ethics committee, ERGO approval number: 106517. Participants gave informed written consent to participate in the grant panel feedback focus group and were separately offered the option to decline inclusion of their transcript in any further research analysis. Data handling, storage, and processing adhered to university policy and GDPR standards. The transcript was fully anonymised prior to analysis. LLM processing was conducted within secure computing environments approved by the host institution.

### Study design, data sources, and participants

We conducted a comparative mixed-methods analysis to evaluate the performance of LLMs and humans in thematic analysis using secondary data consisting of a transcript from a focus group to support the development of an AI-integrated tool to support physical activity recommendations in people living with multiple chronic conditions.

A mixed-methods design was chosen to obtain both quantitative evidence of analytic performance, as well as qualitative insight into areas requiring interpretive nuance. The mixed-methods methodology was chosen to provide methodological triangulation by combining these domains after each was completed to provide an overall synthesis of performance in thematic analysis (Fig 1). This study adheres to GRAMMS (Good Reporting of a Mixed Methods Study) requirements (S1 Text) [20].

**Fig 1. Summary of study protocol.**

The dataset comprised a focus group conducted to explore an AI-informed digital health proposal on interventions in social care and physical activity. The details of this work have been reported previously [21]. Two facilitators led the focus group discussion, which involved a structured presentation of the digital health proposal, followed by discussion on topics including AI interventions in healthcare, care personalisation, and lived experiences. The focus group was conducted in July 2025 and included seven patient and public contributors aged 18 years and older with lived experience of long-term health conditions who were able to provide informed consent. The focus group lasted approximately 2 hours and the transcript length was 12,172 words. The focus group was conducted in line with National Institute of Health Research (NIHR) guidelines for patient and public involvement.

## Data processing and analytic procedures

The dataset was analysed according to a pre-defined analytic framework (S2 Text). The recording of the focus group was transcribed by JK, a physician with experience in qualitative methods and AI implementations in healthcare. The transcript was anonymised at this stage, with any personally identifiable information redacted. No raw identifiable information was processed by any LLM. To ensure comparability and reproducibility, all inputs and outputs for the human and LLM analysis were standardised in JavaScript Object Notation (JSON).

We defined a set of reflexive Braun and Clarke aligned global coding rules to be followed by both the LLM and human parallel analytic streams (S2 Text) [22]. Evidential requirements included the need for both a verbatim quote and segment ID when creating a code. Researcher segments were not coded. Each transcript segment could be used to support one, several, or no codes. For each analytic task, both human qualitative analysts and LLMs were blinded to each other's output.

All LLMs were used according to the pre-defined analytic procedure. Claude 4 Sonnet and ChatGPT-5 were accessed via their respective APIs in October 2025. QualiGPT, a role-based prompt framework that repurposes ChatGPT to conduct systematic inductive and deductive coding, was accessed during the same period through the ChatGPT web interface under identical task specifications [23,24]. Default decoding parameters (temperature = 1.0) were retained to reflect typical, real-world model behaviour and to maintain compatibility across API and web-interface environments. Full prompt templates, JSON formatting, and sample outputs are provided (S2 Text, S1 File, S2 File), with LLM inputs and data processing files publicly available on GitHub [25].

We compared blinded human analysts and LLMs to a reference-standard consensus derived by adjudication. Adjudicated consensus coding was chosen as a pragmatic reference standard, while recognising that thematic interpretation is inherently constructed rather than objectively verifiable. As such, our reference standard represents panel consensus rather than an objective ground truth, enabling a structured evaluation while acknowledging the inherent interpretive nature of thematic analysis.

An expert adjudication panel was purposively constituted with researchers in our multidisciplinary research group, who had established backgrounds in qualitative methods, and knowledge of applying AI in qualitative analysis (CH, GS, AD).

For the deductive analytic task, 10 codes were developed and agreed upon by an expert adjudication group after reviewing the transcript. These codes were grouped into 6 themes. The codes and themes were based on both prevalence, and utility in assessing the LLM's performance in both descriptive and inferential domains of analysis.

The transcript was analysed deductively and reviewed by two researchers (CH and GS). Each participant segment could be labelled either, "true" or "false" in each of the 10 codes, for a total of 1240 data values for each analytic iteration of the transcript. Differences in interpretation were adjudicated by a third researcher (AD) to mediate an expert consensus of the transcript. CH completed an initial coding pass of the transcript according to the predetermined coding rules. GS then performed a second-pass review of the coded transcript and identified discrepant segment-code assignments. Discrepancies were discussed between CH and GS, and unresolved cases were adjudicated by AD, who recorded the final consensus decision with a brief justification. Pre-adjudication agreement was high 99.0% (95% CI 98.4–99.4); κ = 0.931

(95% CI 0.891–0.966); Gwet's AC1 = 0.990 (95% CI 0.984–0.994). Mean prevalence of the codes in the transcript according to the expert panel analysis was 7.8% (SD = 3.2%). The LLMs tested were tasked with analysing the transcript deductively using an identical set of coding rules to the human analysts.

For the inductive task, the expert consensus panel created an adjudicated inductive analysis consisting of 14 codes grouped into 5 themes, reflecting data prevalence and conceptual relevance according to the transcript. Each code was identified according to its label, definition, and supporting evidence.

Both the LLMs and the blinded human analysts were then tasked with performing an identical inductive analysis task according to the pre-defined coding requirements. Strict output formatting in JSON was enforced to enable comparisons between outputs.

Comparative analysis involved comparing the LLMs to humans across the deductive and inductive analytic tasks. The human analysts were blinded to the results of the reference-standard panel analysis and were not involved in drafting the methodology of this study.

We computed and quantified agreement between the LLM outputs and blinded human researchers with the panel human deductive analysis. Agreement was quantified according to several metrics, including percentage agreement, Cohen's Kappa, Gwet's AC1, F1, and Jaccard index [26].

For the comparative inductive analysis, a qualitative researcher compared the LLM outputs to the human inductive analysis. We created a 5-point Likert scale to quantify agreement between the LLM and human analysis (S2 Text), with 5 representing a high level of agreement, and 1 representing poor agreement. The outputs were also reviewed and compared qualitatively to detect differences in interpretation not appreciated by the numerical scale.

The non-inferiority threshold was specified as 0.03 for AC1, reflecting approximate inter-human variation in pilot deductive analysis of the same transcript, representing a pragmatic threshold below which differences would likely reflect typical analytic variability rather than meaningful performance differences. A non-inferiority threshold of 0.5 points on a 5-point Likert scale was chosen for the inductive analysis, representing an acceptable half-step shift in conceptual framing for thematic analysis. Non-inferiority analysis was implemented to test whether model-assisted coding falls within the bounds of expected inter-analyst variation.

Likert scores (1–5) were treated as approximately interval data for calculation of means and confidence intervals. Although Likert responses are ordinal by definition, treating them as approximately interval is common in applied mixed-methods comparisons where the aim is comparative rather than inferential precision. We additionally assessed superiority in each analytic task for the LLMs. We applied a Holm adjustment to control for multiplicity across the three model comparisons [27].

Quote verification involved manually parsing through the LLM inductive analysis to identify whether each segment quote given by the LLM output matches with quotes from the transcript. The reviewing researcher also determined at this stage whether the quote given supports the code assigned by the LLM. Each segment given as evidence for a corresponding code was judged to either be a full match, partial match, or a non-match to the transcript and corresponding code, with non-matching segments counted as a hallucination. Instances where the LLMs coded researcher speech were counted as errors rather than hallucinations.

## Reflexivity

The research team comprised clinicians, qualitative researchers, and researchers with experience in computational methods. The adjudicated panel comprised a senior qualitative researcher (GS), and two other researchers with experience in qualitative research and applications of artificial intelligence (CH, AD). The blinded human analysts included a researcher with experience in qualitative studies and multimorbidity (CP), and a researcher with experience in qualitative research in the context of primary care (DH). All researchers were affiliated with the University of Southampton, working across public health, primary care research, and qualitative analysis. As with all qualitative research, analytic judgements inevitably

PLOS Digital Health

reflect researcher expertise and interpretive perspectives. To reduce reflexive bias, the human analysts were blinded to the background of the study and the LLM outputs. Furthermore, these authors were not involved in development of the evaluation framework or LLM workflows, reducing risk of analytic alignment bias.

## Results

### Comparative deductive analysis

We compared the performance of two LLMs (ChatGPT-5 and Claude 4 Sonnet), and one LLM-based qualitative analysis model (QualiGPT), against two blinded human analysts using human adjudicated analysis, created by an expert consensus adjudication panel of researchers with qualitative backgrounds and experience in AI applications (Table 1). A total of 6200 binary data values were coded in the qualitative analysis: 2480 by human analysts and 3720 by LLMs. Overall agreement was high across all coders. For the 1240 coded data points for each deductive analysis, mean overall agreement was 92.7% (95% CI 91.6-93.9) for the human blinded analysts, and 93.5% (95% CI 92.6-94.5) for the LLMs. Inter-rater reliability expressed as Gwet's AC1 was 0.92 (95% CI 0.90–0.93) for human analysts and 0.93 (95% CI 0.92–0.94) for the LLMs. Values of Cohen's κ were lower with mean values of 0.34 for both groups, representing the attenuation of κ when applied to low-prevalence datasets. Jaccard index was computed as a measure of overlap of positive codes within the dataset and was similar for both the human and LLM analyses.

Non-inferiority and superiority testing (Fig 2A) were performed in the deductive analysis according to a Holm-adjusted, non-parametric, one-sided bootstrap test with segment-level resampling of the difference in AC1 between the LLM results and the mean of human analysts. All LLMs were non-inferior to human analysts ($p < 0.0001$) relative to the predetermined non-inferiority value (0.03). ChatGPT-5 ($p = 0.048$) and Claude 4 Sonnet ($p = 0.039$) achieved statistical superiority over human analysts in the deductive analysis.

Measures of coding accuracy included sensitivity, specificity, and F1, with comparable results between the blinded human analysts and LLMs according to each measure. The assessed LLMs demonstrated a high specificity (0.98, 95% CI 0.97-0.98) when tested on the focus group transcript, suggesting that the LLMs are inclined to interpret the transcript conservatively, rarely assigning codes unless clear textual evidence is present. Sensitivity was lower for both groups, reflecting the challenge of consistently identifying infrequent or implicit themes within the data.

Thematic-level agreement analysis (Table 2) demonstrated variable performance across the codes tested, with Cohen's κ ranging from –0.05 to 0.54. Human blinded analysts achieved the highest mean concordance within the codes of Privacy and Data Concerns, Consent and Transparency, and Need for personalisation. LLMs demonstrated highest concordance in the codes of Human and AI Interactions in Healthcare, Privacy and Data Concerns, and Consent and Transparency. Lower concordance was observed in themes requiring interpersonal or inferential reasoning such as Patient

**Table 1. Overall agreement vs. reference-standard human adjudicated panel, expressed as mean values with 95% bootstrap confidence intervals (CIs). Values are reported as number (95% CI) for each metric: Agreement (%), Cohen's κ, Gwet's AC1, Jaccard index, Sensitivity, Specificity. All metrics shown with 95% bootstrap CIs (B = 1000).**

| Model/Coder | Agreement % | Cohen's κ | Gwet's AC1 | Jaccard | Sensitivity | Specificity | F1 |
|---|---|---|---|---|---|---|---|
| Human Analyst 1 | 91.7 (90.4–92.9) | 0.25 (0.17–0.34) | 0.91 (0.89–0.92) | 0.18 (0.12–0.23) | 0.28 (0.21–0.38) | 0.96 (0.95–0.97) | 0.30 (0.22–0.38) |
| Human Analyst 2 | 93.8 (92.7–94.8) | 0.42 (0.32–0.51) | 0.93 (0.92–0.94) | 0.29 (0.22–0.36) | 0.41 (0.31–0.50) | 0.97 (0.97–0.98) | 0.45 (0.36–0.53) |
| Human mean | 92.7 (91.6–93.9) | 0.34 (0.26–0.41) | 0.92 (0.90–0.93) | 0.23 (0.18–0.29) | 0.34 (0.27–0.42) | 0.97 (0.96–0.97) | 0.37 (0.30–0.45) |
| ChatGPT-5 | 93.5 (92.3–94.7) | 0.31 (0.22–0.39) | 0.93 (0.92–0.94) | 0.21 (0.14–0.27) | 0.27 (0.19–0.34) | 0.98 (0.97–0.99) | 0.34 (0.26–0.43) |
| Claude 4 Sonnet | 93.8 (92.7–94.9) | 0.35 (0.26–0.44) | 0.93 (0.92–0.94) | 0.24 (0.18–0.31) | 0.31 (0.23–0.39) | 0.98 (0.97–0.99) | 0.39 (0.29–0.47) |
| QualiGPT | 93.3 (92.1–94.4) | 0.34 (0.25–0.43) | 0.93 (0.91–0.94) | 0.23 (0.17–0.30) | 0.33 (0.25–0.41) | 0.97 (0.97–0.98) | 0.38 (0.29–0.46) |
| LLM mean | 93.5 (92.6–94.5) | 0.34 (0.26–0.40) | 0.93 (0.92–0.94) | 0.23 (0.17–0.28) | 0.30 (0.24–0.36) | 0.98 (0.97–0.98) | 0.37 (0.29–0.43) |

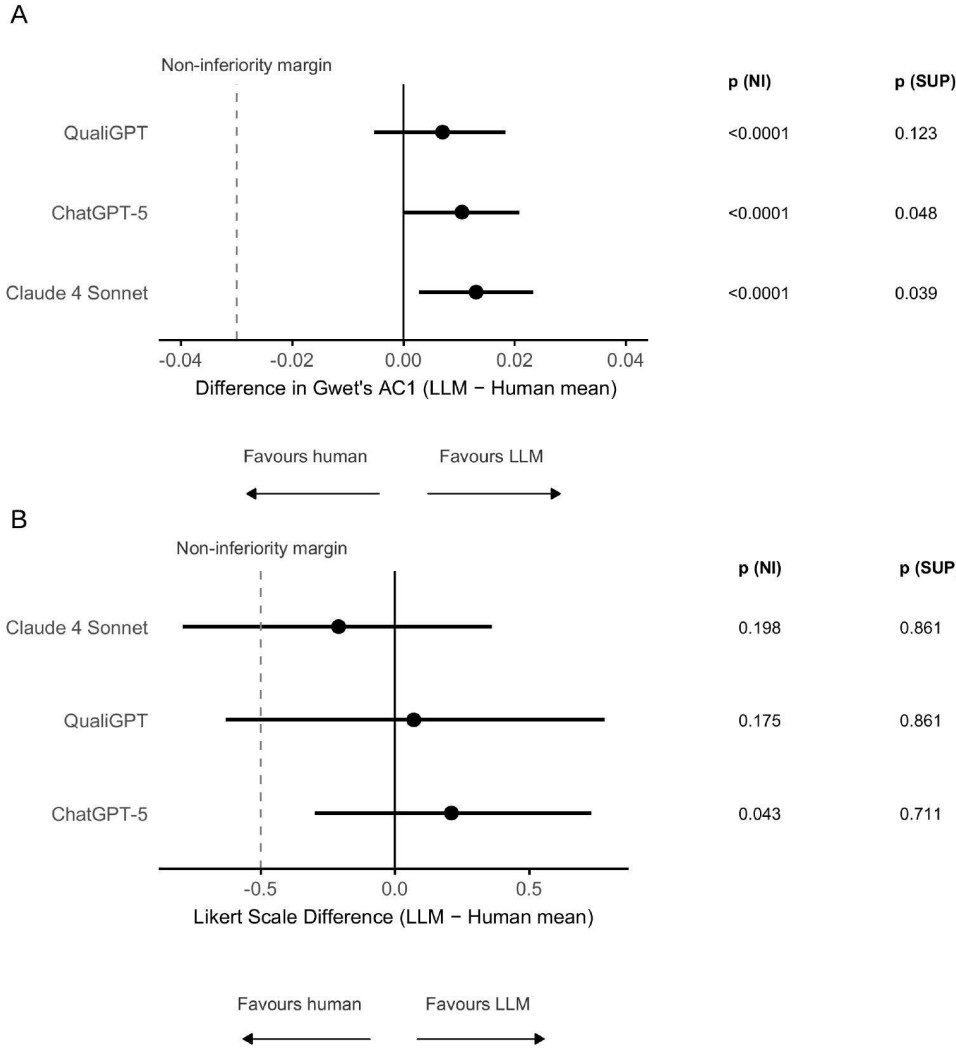

**Fig 2. (A) Differences in agreement for deductive coding, expressed as the difference in AC1 between each LLM and the mean of blinded human analysts.** The vertical dashed line indicates the prespecified non-inferiority margin (–0.03). Non-inferiority and superiority were tested using a Holm-adjusted, one-sided bootstrap test of the difference in AC1. **(B)** Differences in inductive thematic analysis scores, expressed as the Likert-scale difference (5-point scale) between each LLM and the mean of blinded human analysts. The dashed line indicates the prespecified non-inferiority margin (–0.5). Non-inferiority and superiority were assessed using a Holm-adjusted, one-sided t-test. Abbreviations: **NI** = non-inferiority, **SUP** = superiority, error bars represent 95% CIs derived from bootstrap resampling **(A)** and parametric variance estimation **(B)**.

Expertise and Collaborative Involvement of Patients in Design. Despite these differences, concordance was similar between human analysts and LLMs, indicating broadly similar concordance across all codes.

## Comparative inductive analysis

The comparative inductive analysis (Table 3 and S3 File) was conducted by an expert consensus adjudication group. Agreement was assessed qualitatively and using a 5-point Likert scale (S2 Text), where 5 indicates complete alignment. Performance of LLMs and human analysts on the Likert scale was comparable. Only one LLM (ChatGPT-5) reached non-inferiority at the pre-specified margin (Fig 2B). We observed the highest level of congruence in descriptive themes

**Table 2. Cohen's kappa (κ) by theme compared with the reference-standard human adjudicated panel, with 95% bootstrap confidence intervals (B = 1000).**

| Code | Human Analyst 1 | Human Analyst 2 | ChatGPT-5 | Claude 4 Sonnet | QualiGPT | Human mean | LLM mean |
|---|---|---|---|---|---|---|---|
| Co-production: Collaborative involvement | 0.01 (-0.11–0.17) | 0.29 (0.06–0.51) | 0.05 (-0.08–0.24) | -0.05 (-0.08—0.02) | 0.07 (-0.07–0.26) | 0.15 (-0.01–0.33) | 0.02 (-0.06–0.13) |
| Data Security and Ethics in AI: Consent and transparency | 0.27 (-0.03–0.61) | 0.78 (0.49–1.00) | 0.46 (-0.01–0.80) | 0.58 (0.20–0.89) | 0.22 (-0.04–0.53) | 0.52 (0.32–0.71) | 0.42 (0.07–0.68) |
| Data Security and Ethics in AI: Privacy and data concerns | 0.57 (0.18–0.85) | 0.69 (0.27–1.00) | 0.46 (-0.01–0.79) | 0.42 (-0.01–0.75) | 0.46 (-0.02–0.79) | 0.64 (0.22–0.93) | 0.45 (0.08–0.74) |
| Group Cohesion: Positive peer support | 0.31 (-0.02–0.64) | 0.32 (-0.01–0.61) | 0.22 (-0.03–0.49) | 0.41 (0.11–0.70) | 0.25 (-0.04–0.53) | 0.31 (-0.02–0.58) | 0.29 (0.04–0.53) |
| Perceptions of the Health System: Care fragmentation | 0.46 (-0.02–0.82) | 0.47 (-0.01–0.82) | 0.43 (-0.01–0.79) | 0.54 (0.17–0.85) | 0.22 (-0.04–0.53) | 0.47 (0.10–0.75) | 0.39 (0.12–0.63) |
| Perceptions of the Health System: Trust/mistrust | 0.05 (-0.09–0.24) | 0.49 (0.24–0.71) | 0.23 (-0.03–0.48) | 0.35 (0.07–0.61) | 0.39 (0.10–0.66) | 0.27 (0.10–0.44) | 0.32 (0.11–0.53) |
| AI in Healthcare: Human v AI interactions in healthcare | 0.00 (-0.09–0.17) | 0.22 (-0.02–0.52) | 0.46 (0.17–0.72) | 0.54 (0.24–0.79) | 0.50 (0.20–0.77) | 0.11 (-0.03–0.27) | 0.50 (0.23–0.71) |
| AI in Healthcare: System integration | 0.24 (-0.04–0.51) | 0.27 (-0.03–0.52) | 0.37 (0.00–0.66) | 0.23 (-0.03–0.53) | 0.35 (-0.01–0.66) | 0.26 (0.04–0.48) | 0.31 (0.06–0.57) |
| Tailoring Care for Multimorbidity: Need for personalisation | 0.53 (0.28–0.75) | 0.46 (0.23–0.68) | 0.38 (0.12–0.62) | 0.40 (0.15–0.61) | 0.46 (0.24–0.67) | 0.50 (0.27–0.70) | 0.41 (0.23–0.58) |
| Tailoring Care for Multimorbidity: Patient expertise | 0.33 (0.05–0.61) | 0.28 (0.03–0.53) | 0.19 (-0.04–0.47) | 0.19 (-0.04–0.48) | 0.46 (0.12–0.76) | 0.31 (0.09–0.52) | 0.28 (0.04–0.54) |

**Table 3. Mean agreement with the reference-standard human adjudicated panel across inductive analyses for human analysts 1 and 2, and the three assessed LLMs.**

| Human Adjudicated Code | Human Analyst 1 | Human Analyst 2 | ChatGPT-5 | Claude 4 Sonnet | QualiGPT |
|---|---|---|---|---|---|
| Desire for personalisation | 4 | 5 | 4 | 3 | 3 |
| Truth and misinformation | 4 | 5 | 4 | 4 | 4 |
| Harm from generic advice | 5 | 4 | 4 | 4 | 1 |
| Need for flexibility and adaption | 3 | 3 | 5 | 5 | 5 |
| NHS lack of personalisation | 2 | 4 | 4 | 3 | 5 |
| Difficulty accessing healthcare | 4 | 5 | 5 | 5 | 5 |
| Care continuity/fragmentation | 5 | 3 | 5 | 1 | 4 |
| Desire for personalisation in AI interactions | 3 | 4 | 4 | 3 | 3 |
| Need for human and social interaction | 4 | 4 | 2 | 4 | 5 |
| Opinions of communities and peer support | 3 | 4 | 4 | 5 | 3 |
| Feedback on AI development | 3 | 4 | 3 | 3 | 5 |
| Scepticism surrounding security | 4 | 5 | 4 | 5 | 5 |
| Scepticism of medical experts | 4 | 2 | 5 | 2 | 4 |
| Peer support | 3 | 5 | 4 | 4 | 3 |
| **Mean (95% CI)** | **3.643 (3.157–4.129)** | **4.071 (3.542–4.601)** | **4.071 (3.593–4.550)** | **3.643 (2.941–4.345)** | **3.929 (3.232–4.625)** |

expressed in the transcript. These included the need for personalisation, difficulty accessing healthcare, harm from generic advice, and scepticism surrounding data security.

Across both human analysts and LLMs, a high degree of overall thematic convergence emerged. Analysts identified speech segments such as the following:

*T037: "We as individuals monitor how we respond. I mean, there are at least three of us in this room".*

This was universally detected as reflecting the desire for individual treatment in the context of healthcare and multimorbidity. Although it was sometimes framed in analogous or overlapping codes such as "individualised treatment needs", the underlying emphasis on personalisation was preserved between analysts.

Significant agreement was identified in the theme of Truth, misinformation, and guidance. For example:

T002: *"you can often read things and then you do a little bit of research around it and you find that actually the truth is perhaps even the opposite"*

This statement was widely recognised as an expression of misinformation and trust degradation in everyday healthcare interactions.

While there was strong overall agreement in thematic identification, key divergences emerged in how human analysts and LLMs conceptualised and framed these themes. Human analysts tended towards a relational conceptualisation of personalisation with codes such as "personable approach", and "patients as individuals". Although both human analysts and LLMs identified the need for personalisation in healthcare, LLMs tended to frame the lack of such personalisation as a systemic design issue needing to be addressed, while human analysts framed the problem at a more personal, affective level.

Despite the expert adjudicated panel agreeing that the theme of care fragmentation featured prominently in the transcript, there was variation in how well it was addressed, specifically by the LLMs. Although accessibility barriers were frequently commented upon, the code of care continuity and fragmentation was an area of variable LLM performance. We noted this domain as a specific example of weakness in inferential reasoning, with the LLMs seemingly struggling to conceptualise the speech within this code.

Peer support was another theme that illustrated divergence. All analysts recognised it as a significant social factor, but the depth of interpretation differed. Human analysts demonstrated stronger inferential capacity in this domain, while the LLMs seemingly struggled to infer peer support and connection beyond what was explicitly stated in the text. For example:

*T040: "The community aspect of it, about people being able to connect with people locally, maybe with the same condition or maybe just with the same exercise routines."*

This was widely identified by the LLMs as an explicit example of community support. This theme, relying heavily on inferential and latent meaning, was a domain in which human analysts added significant additional insight, while the LLMs produced a much more descriptive analysis.

The LLMs contributed alternative analytical insights by framing participants' experiences through a more systemic and process-oriented lens than human analysts. Whereas human coders tended to interpret negative experiences as rooted in structural or interpersonal factors, the LLMs linked poor experience more systemically to outdated systems, training, and guidance. Interestingly, AI implementations featured more heavily in the inductive analysis by the LLMs, even when speech segments seemingly had less relation to AI.

Trust and feedback in AI development had variable interpretations. The LLMs diverged significantly in their analysis, with QualiGPT particularly focusing on language use within AI implementations. Claude 4 Sonnet coded "AI learning

potential", with some overlap in thematic analysis. LLM codes such as "compliance language" reflected their tendency to focus on practical tangible implementations and improvements.

Further evidence of additional insight appeared in the domain of flexibility and adaptation. For example:

*T042: "For fluctuating conditions, can it change like that?"*

This quote was more readily translated by the LLMs into specific design requirements, coded as "need for flexible, adaptive programs." Human analysts, by contrast, differed in coding this segment, with only one analyst identifying it distinctly. This suggests that LLMs may exhibit a comparative advantage in abstracting technical or system-level implications from participants' remarks, whereas human analysts prioritised the subjective, experiential meaning of those same expressions.

### Result verification

We quantified the accuracy and validity of the LLM coding according to the metrics of strict hallucination rate, expanded hallucination rate, and comprehensive error rate (Table 4). The strict hallucination rate included segments which did not support the code assigned by the LLM, aligning with the definition of hallucination in machine learning. The overall hallucination rate of the models tested was 1.2% (SD = 2.1%), with two out of the three models tested having no hallucinations in the inductive analysis. The expanded hallucination rate included both segments identified as erroneous evidence, and researcher segments coded in the inductive analysis. The mean expanded hallucination rate was 8.6% (SD = 5.1%) for the models tested, significantly higher than the strict definition of hallucination. Finally, the comprehensive error rate included partial matches of segment-code meaning in its definition. The mean comprehensive error rate was 12.4% (SD = 5.1%) for the models tested.

We report several metrics for thorough evaluation of error rates in qualitative analysis. It should be noted that not all error types have equivalent implications for qualitative validity. Segments classified as strict hallucinations represent unsupported evidence and therefore pose the greatest risk to analytic validity. In contrast, researcher segments coded typically reflect instruction-following or speaker attribution failures, reflecting on the practical ability of the LLM to interpret the coding task. Partial matches usually indicate borderline or differing interpretation rather than fabrication of evidence.

**Table 4. LLM quote verification and overall error rates for each LLM.**

| Model Coding Summary (per-model: n and %; mean (SD)) | | | | |
|---|---|---|---|---|
| **Metric** | **ChatGPT-5** | **QualiGPT** | **Claude 4 Sonnet** | **Mean (SD)** |
| Exact match | 31 (91.2%) | 45 (81.8%) | 71 (89.9%) | 49.0 (20.3) |
| Partial Match | 2 (5.9%) | 3 (5.5%) | 0 (0.0%) | 1.7 (1.5) |
| No Match | 0 (0.0%) | 2 (3.6%) | 0 (0.0%) | 0.7 (1.2) |
| Researcher segments coded | 1 (2.9%) | 5 (9.1%) | 8 (10.1%) | 4.7 (3.5) |
| **Total Segments Coded** | **34 (100%)** | **55 (100%)** | **79 (100%)** | **56.0 (22.5)** |
| **Overall Error Rates** | | | | |
| Strict Hallucination Rate (No Match/ Total) | 0/34 (0.0%) | 2/55 (3.6%) | 0/79 (0.0%) | 1.2% (2.1%) |
| Expanded Hallucination Rate (incl. policy violations) | 1/34 (2.9%) | 7/55 (12.7%) | 8/79 (10.1%) | 8.6% (5.1%) |
| Comprehensive Error Rate (incl. partial matches) | 3/34 (8.8%) | 10/55 (18.2%) | 8/79 (10.1%) | 12.4% (5.1%) |

Percentages in the main block are column-wise (denominator = each model's Total Segments Coded).

Strict = No Match/Total. Expanded = (No Match + Researcher-segments-coded)/Total. Comprehensive = (No Match + Researcher-segments-coded + Partial Match)/Total.

Fourth column: Mean(SD) of counts (main block) and of percentages (error-rate block).

Comprehensive error rates should not be interpreted as comprising uniformly consequential errors, but as representing a spectrum of potential issues when applying LLMs to qualitative analysis.

## Discussion

In this mixed-methods comparison, we observed comparable performance between LLMs and blinded human analysts. The strengths of the LLMs included their application to descriptive qualitative codes, as well as their high specificity in coding the transcript. Our dataset, reflective of a practical, unrefined focus group transcript, had a low overall prevalence of qualitative codes, enabling evaluation performance under minimally curated conditions. It is well-documented that Cohen's κ and Jaccard index are significantly affected by the prevalence within a sample [28,29]. Although we report high raw agreement values, the "prevalence effect" of such metrics leads to lower agreement metrics than have been reported elsewhere. Comparative analysis according to deductive codes and themes did not yield significant differences between humans and models tested; although we observed a trend toward stronger model performance in descriptive coding.

Our results introduce a novel methodological approach to the comparison of humans and LLMs in thematic analysis. Each of the models tested performed well as defined by strict hallucinations, with most models outputting no erroneously coded sections. However, performance declined in speaker delineation according to the inductive analysis prompt, despite researcher speech segments being clearly labelled in the transcript. Although the strict hallucination rate of the LLMs was relatively low (1.2%), the comprehensive error rate of 12.4% was considerably higher. This has significant implications as to the utility of LLMs in qualitative analysis of focus groups and arose largely from the LLMs coding facilitator speech segments in the transcript. Although researcher speech segments could be removed in data pre-processing, it could be argued that this removes part of the transcript data which may have implications for its overall meaning.

This finding highlights a broader consideration in applying LLMs to qualitative data: while pre-processing can improve technical performance metrics, it also risks undermining context important for interpretive validity. Furthermore, the use case for LLMs within qualitative research necessitates the ability of such models to accurately perform predetermined instructions and adhere to coding requirements set by researchers. Our findings illustrate an epistemic issue in applying LLMs to coding tasks: LLMs do not "understand" coding instructions as a human analyst but approximate them using probabilistic pattern-recognition.

Our comparative inductive analysis aligns with prior research suggesting that LLMs can reproduce prominent themes and sometimes expand upon or offer additional insights to human coded data [30–32]. Prior research has questioned the ability of LLMs to detect latent meaning and affective responses within qualitative datasets [33–35]. Comparative inductive analysis demonstrated weaknesses in domains such as peer support and co-production. Further research could explore this in more detail.

Our results affirm the emerging consensus that LLMs perform best in constrained, deductive coding tasks, and less effectively in inductive analysis requiring sensitivity to interpersonal nuance and latent meaning [2,36]. This mirrors findings in broader research suggesting that human-AI hybrid implementations are best placed to balance efficiency with accuracy and validity [37,38].

Prior methodologies have measured agreement between LLMs and human in qualitative analysis using quantitative agreement metrics and Likert-scale agreement [4,39]. Deiner et al., 2024 provides a methodological treatment of hallucinations when applying LLMs to thematic analysis [30]. While our findings similarly indicate low overall hallucination rates, we extend this framework by introducing a comprehensive error rate. Furthermore, although previous studies have reported the use of blinded analysts, to our knowledge, no studies have applied a formal non-inferiority framework. As such, our study assesses functional comparability rather than descriptive similarity alone.

Moreover, LLMs may offer a reduction in time required for initial coding and analysis of large datasets. However, this benefit must be balanced against the time required for verification, ensuring accuracy and appropriate data interpretation. Once again, the net efficiency of LLM integration will depend on the nature of its utilisation. We did not formally assess

time spent on each task by human analysts, or time for verification of LLM outputs. As such, we are not able to quantify efficiency or cost-effectiveness of LLM-assisted workflows. Future work would quantify this in more detail.

It is suggested from our results that LLMs framed many participant experiences through a process-oriented, systems level lens, whereas human analysts emphasised lived experience and emotional nuance. This may reflect the orientation of LLMs towards academic and technical discourse, subsequently re-contextualising personal narratives through procedural frameworks. On one hand, emotional depth may be attenuated; however, in policy driven research, such a perspective may be advantageous. Therefore, the effects on the framing should be interpreted contextually.

## Strengths and limitations

This study's strengths include a blinded mixed-methods design to enable a robust comparison of model performance compared to qualitative analysts across a variety of metrics. By integrating quantitative agreement metrics with qualitative interpretation, our study offers a reproducible framework for evaluating AI performance in thematic analysis. We utilised a minimally curated focus group dataset to ensure an accurate assessment of the practical performance of LLMs in thematic analysis, while applying a uniform analytic framework which allowed for direct comparison between the LLMs and human analysts. Additionally, we tested the models according to both deductive and inductive qualitative analysis, while also reporting hallucination and error rates of the LLMs tested. Finally, this study offers methodological transparency through the open sharing of prompts, task specifications, and LLM outputs.

However, several limitations should be acknowledged. The study was based on a single focus group transcript with seven participants discussing the topic of AI implementations in social care and physical activity. Subsequently, findings should therefore be interpreted as having context-dependent generalisability. The use of AI in healthcare is a widely discussed, policy-oriented subject, and thus may align well with the training of LLMs. As such, the LLM outputs may reflect patterns from existing discourse as opposed to novel data synthesis.

We would therefore expect model-human comparability to differ across topics. In domains that are highly represented in public and policy discourse (e.g., AI in healthcare), LLMs may more readily reproduce common framings and terminology, potentially inflating apparent alignment. Conversely, for topics that are less represented in training data, more culturally specific, or more emotionally complex, models may show reduced sensitivity to locally grounded meaning and greater reliance on generic or 'template' interpretations.

Our findings are also model and version-specific and may shift according to future model updates. It should be noted that the adjudicated human reference-standard panel which acted as the comparator for the primary findings of this study was the interpretation of a panel of researchers, and not an objective ground truth. As such we evaluated whether the outputs were aligned with this panel-consensus, rather than if they were epistemically correct.

Our non-inferiority analysis treated two blinded human analysts as a benchmark for human performance. Given the interpretive nature of qualitative analysis, this benchmark is subject to intrinsic variation according to the skills, style, and interpretive depth of the analysts chosen. The human comparator in our study comprised two researchers from a single institution, limiting the ability to characterise variability across human analysts more broadly. We also note that the scope of non-inferiority testing was confined to two specified analytic tasks applied to a single focus group dataset.

Finally, limitations in the statistical analysis of the data should also be acknowledged. Our deductive analysis produced narrow confidence intervals, in part due to the larger number of data points, and significant effect separation from the pre-specified margin. However, our comparative inductive analysis involved fewer codes, reducing the statistical power to detect a difference at the pre-specified non-inferiority margin; inductive comparisons should therefore be interpreted cautiously. Further studies should develop multi-group sampling to obtain stable estimates of inductive analysis performance.

 

## Conclusions

This study introduces a reproducible and blinded evaluation framework for comparing large language models (LLMs) and human researchers in thematic analysis of healthcare data. Through structured data formatting and analysis, we demonstrate how computationally rigorous approaches can complement qualitative interpretation

We suggest several methodological contributions for future consideration in mixed-methods research on the application of LLMs to qualitative data. The use of global and local coding rules with strict output formatting requirements improved the auditability of the comparison. The use of blinded human researchers further reduced reflexivity and expectancy bias, allowing for valid comparisons between human analysts and LLMs. Quote verification provided a grounded view of error modalities when applying LLMs to qualitative data. Finally, by testing the LLMs across both inductive and deductive analytic tasks, we identified variation in their interpretive capacity while also quantifying their relative performance.

From a clinical informatics perspective, these methodological refinements support the responsible integration of LLMs into health research workflows. The implementation of transparent and reliable qualitative analysis underpins patient and clinician-centred design of decision-support tools, healthcare implementations, and policy frameworks. LLMs that can deliver reproducible qualitative analysis while maintaining low error rates have the potential to accelerate the synthesis of patient and clinician insight into system design and clinical decision-making frameworks, while preserving the necessity of human oversight.

LLMs may be perceived as assistive analytic tools for refinement rather than a replacement for human judgement. Their effectiveness will vary depending on the context being explored. Topics involving personal, or nuanced experiences may require human-led analysis, whereas well-documented technical topics may benefit from LLM intervention. We propose the following practical decision framework for LLM use in thematic analysis:

(i)   LLM assistance is favoured when the goal is large-volume descriptive coding, or mapping text to a pre-defined framework

(ii)  Human interpretation is favoured when the analytic goal or framework is deeply interpretive

(iii) If LLMs are used in qualitative analysis, we suggest quote verification and reporting of error rates, with documentation on how model outputs are integrated into final analysis

This study's findings highlight an opportunity for the development of specialised, clinically contextualised LLMs trained to capture the relational and affective dimensions of patient narratives. Whereas general-purpose models perform well in descriptive or procedural coding, healthcare decision-making frequently depends on understanding empathy, trust, and interpersonal meaning, domains where current models demonstrate limitations. Future research should explore the fine-tuning of LLMs on clinical datasets, enabling the emergence of relationally competent LLMs capable of detecting trust, affect, and empathy.

Collectively, this study introduces a reproducible, blinded evaluation framework for comparing LLM and human performance in thematic analysis in a healthcare setting, incorporating formal non-inferiority testing and error auditing. While LLMs demonstrated non-inferior performance in deductive analysis of a polyvocal transcript, their higher comprehensive error rate and weaker performance in affective coding supports their use as an adjunct, rather than a replacement for human qualitative analysts.

## Supporting information

**S1 Text. GRAMMS checklist.**
(DOCX)

**S2 Text. Study protocol.**
(PDF)

**S1 File. Deductive results.**
(XLSX)

**S2 File. Inductive results.**
(TXT)

**S3 File. Comparative inductive analysis.**
(XLSX)

## Acknowledgments

We would like to thank all the patients and participants who contributed to this work, and also to Lucy Smith for leading the original focus group.

## Author contributions

**Conceptualization:** Callum Hill, Arun Dahil, Glenn Simpson, Hajira Dambha-Miller.

**Data curation:** Callum Hill, Arun Dahil.

**Formal analysis:** Callum Hill, Arun Dahil, Glenn Simpson, David Hardisty, Cameron Kumar Pinn.

**Funding acquisition:** Hajira Dambha-Miller.

**Investigation:** Callum Hill, Arun Dahil, Hajira Dambha-Miller.

**Methodology:** Callum Hill, Glenn Simpson, Hajira Dambha-Miller.

**Project administration:** Callum Hill, Hajira Dambha-Miller.

**Resources:** Hajira Dambha-Miller.

**Software:** Callum Hill.

**Supervision:** Hajira Dambha-Miller.

**Visualization:** Callum Hill.

**Writing – original draft:** Callum Hill, Arun Dahil, Hajira Dambha-Miller.

**Writing – review & editing:** Callum Hill, Glenn Simpson, David Hardisty, Jacob Keast, Cameron Kumar Pinn, Hajira Dambha-Miller.

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
