## [Decision Letter · Decision Letter 0]

9 Feb 2026

Response to Reviewers '. This file does not need to include responses to any formatting updates and technical items listed in the 'Journal Requirements' section below.* A marked-up copy of your manuscript that highlights changes made to the original version. You should upload this as a separate file labeled 'Revised Manuscript with Track Changes '.* An unmarked version of your revised paper without tracked changes. You should upload this as a separate file labeled 'Manuscript '. If you would like to make changes to your financial disclosure, competing interests statement, or data availability statement, please make these updates within the submission form at the time of resubmission. Guidelines for resubmitting your figure files are available below the reviewer comments at the end of this letter. We look forward to receiving your revised manuscript. Kind regards, Michael WinterAcademic EditorPLOS Digital Health Michael WinterAcademic EditorPLOS Digital Health Leo Anthony CeliEditor-in-ChiefPLOS Digital Healthorcid.org/0000-0001-6712-6626  **Journal Requirements:** If the reviewer comments include a recommendation to cite specific previously published works, please review and evaluate these publications to determine whether they are relevant and should be cited. There is no requirement to cite these works unless the editor has indicated otherwise.  **Additional Editor Comments (if provided):****Reviewers' Comments:** Reviewer's Responses to Questions

**Comments to the Author**

1. Does this manuscript meet PLOS Digital Health’s publication criteria ? Is the manuscript technically sound, and do the data support the conclusions? The manuscript must describe methodologically and ethically rigorous research with conclusions that are appropriately drawn based on the data presented.

Reviewer #1: Yes

Reviewer #2: Yes

Reviewer #3: Yes

2. Has the statistical analysis been performed appropriately and rigorously?

Reviewer #1: Yes

Reviewer #2: Yes

Reviewer #3: Yes

3. Have the authors made all data underlying the findings in their manuscript fully available (please refer to the Data Availability Statement at the start of the manuscript PDF file)?

Reviewer #1: Yes

Reviewer #2: Yes

Reviewer #3: Yes

4. Is the manuscript presented in an intelligible fashion and written in standard English?

Reviewer #1: Yes

Reviewer #2: Yes

Reviewer #3: Yes

Reviewer #1: This study evaluates the performance of large language models (LLMs) in qualitative thematic analysis of healthcare focus-group data using a blinded mixed-methods framework. Two general-purpose LLMs and an LLM-based coding tool were compared with blinded human analysts across deductive and inductive coding tasks, using expert-adjudicated reference standards, agreement metrics, non-inferiority testing, and error analysis. Results show that LLMs are non-inferior to human analysts in deductive coding but exhibit more variable performance in inductive analysis, with low hallucination yet notable comprehensive error rates. The findings demonstrate that LLMs can augment, but not replace, human qualitative analysis and introduce a reproducible framework for evaluating AI-assisted qualitative research.

This is a strong and well-written manuscript. It can be improved by attending to the following:

1-Introduction: In situating the proposed research within prior work, consider providing a thematic or theoretical synthesis beyond a descriptive synthesis.

2-Methodology: Consider strengthening the justification of methodological choices, e.g., the epistemological assumptions behind the use of expert adjudication as a reference model, non-inferiority margins, and the treatment of Likert-scale as interval data.

Reviewer #2: The authors identify a gap in our knowledge which they sought to fill, "How good are LLMs at analysing transcripts for qualitative, thematic analysis?" They used a solid methodology comparing multiple LLM and humans to reference data, using a dataset of historical focus group sessions. They find that LLMs are as good as humans, albeit with some blind spots and differences of emphasis. They conclude LLMs can be used to support human analysts rather than replacing them. They propose development of specialised LLMs for thematic analysis of focus groups.

The paper is well written and easy to read.

Methodological Critique

The methodology compares coding from three LLMs and two human analysts to a "gold-standard" coding provided by an adjudication panel. They were each provided with transcripts of historical sessions.

The results crucially depend on the human transcriber and adjudication panel. It took me a little while to realize that these were four authors. This is not made explicit in the report, only implied by the use of parenthetical initials. I recommend the authors roles in the methodology be made more clear in the report, also clarifying their division of expertise and diversity across institutions. This is a potential important methodological weakness so the paper would benefit from discussing possible biases and conflicts of interest.

With only two human analysts in the methodology, it is difficult to appreciate the variation across humans. So when the authors say things like "Human analysts [...] showed less consensus in coding this segment" it is not clear that the methodology can support such observations.

Limitations of the methodology are mentioned in "Strengths and limitations" but how they may affect interpretation of the results is not discussed.

Despite these methodological questions, the conclusions of the study seem trustworthy and the paper makes a good contribution.

Suggestions for Minor Improvements

It would be helpful to characterise the dataset of focus groups and sessions. Only one is discussed in "Materials and Methods" and the nature of the inputs is not clear to me.

Make a clearer characterisation of the adjudication panel.

It would be helpful to define PPI the first time it is used in the prose.

Figure 1 is difficult for me to read as the quality of the image is low. The text is small and slightly blurred.

Page 8, line 141 contains a typo, "streams (S2 text)t."

Reviewer #3: This is an interesting, well-designed mixed-methods comparison. With the increasing use LLMs and human analysts working in conjuction in analysis on healthcare focus group data, this is a timely and important study. A few questions and suggestions:

1. The reliance on a single focus group with seven participants discussing one specific topic (AI in healthcare) significantly limits generalizability.

a. Can the authors more explicitly discuss how topic familiarity might influence results, would you expect the results to be different as below?

b. Consider whether LLM performance might differ when analyzing discussions about topics less represented in training data

2. The authors acknowledge the adjudication panel represents researcher interpretation rather than "objective truth." However:

a. Could you provide more detail on the adjudication process? How were disagreements resolved?

What was the initial inter-rater reliability between the two researchers (CH and GS) before adjudication by AD?

3. Table 4: The comprehensive error rate definition includes three components, but their relative importance isn't clear. Could you provide a weighted error metric or discuss whether all errors are equally consequential for qualitative analysis?

4. LLMs "framed participants' experiences through a process-oriented lens" and featured AI more prominently is a interesting perspective. Is this a strength or weakness (missing lived experience)? Does this represent inherent model bias or (in)appropriate technical framing?

5. Consider adding information on when LLMs should vs. shouldn't be used in qualitative analysis, perhaps with a decision framework for researchers.

6. What about practical adoption? cost-effectiveness (time saved vs. verification effort required) would be valuable for researchers considering LLM adoption.

**Do you want your identity to be public for this peer review?** For information about this choice, including consent withdrawal, please see our Privacy Policy .

Reviewer #1: No

Reviewer #2: No

Reviewer #3: No

**Figure resubmission:**  While revising your submission, we strongly recommend that you use PLOS’s NAAS tool (https://ngplosjournals.pagemajik.ai/artanalysis) to test your figure files. NAAS can convert your figure files to the TIFF file type and meet basic requirements (such as print size, resolution), or provide you with a report on issues that do not meet our requirements and that NAAS cannot fix.

**Reproducibility:** To enhance the reproducibility of your results, we recommend that authors of applicable studies deposit laboratory protocols in protocols.io, where a protocol can be assigned its own identifier (DOI) such that it can be cited independently in the future. Additionally, PLOS ONE offers an option to publish peer-reviewed clinical study protocols. Read more information on sharing protocols at https://plos.org/protocols?utm_medium=editorial-email&utm_source=authorletters&utm_campaign=protocols To enhance the reproducibility of your results, we recommend that authors of applicable studies deposit laboratory protocols in protocols.io, where a protocol can be assigned its own identifier (DOI) such that it can be cited independently in the future. Additionally, PLOS ONE offers an option to publish peer-reviewed clinical study protocols. Read more information on sharing protocols at https://plos.org/protocols?utm_medium=editorial-email&utm_source=authorletters&utm_campaign=protocols

---

## [Decision Letter · Decision Letter 1]

6 Mar 2026

Large Language Models for Thematic Analysis in Healthcare Research: A Blinded Mixed-Methods Comparison with Human Analysts

PDIG-D-25-01248R1

Dear Mr Hill,

We are pleased to inform you that your manuscript 'Large Language Models for Thematic Analysis in Healthcare Research: A Blinded Mixed-Methods Comparison with Human Analysts' has been provisionally accepted for publication in PLOS Digital Health.

Best regards,

Michael Winter

Academic Editor

PLOS Digital Health

**Additional Editor Comments (if provided):**

**Reviewer Comments (if any, and for reference):**

Reviewer's Responses to Questions

**Comments to the Author**

Reviewer #1: All comments have been addressed

Reviewer #2: All comments have been addressed

Reviewer #3: All comments have been addressed

publication criteria ? Is the manuscript technically sound, and do the data support the conclusions? The manuscript must describe methodologically and ethically rigorous research with conclusions that are appropriately drawn based on the data presented.

Reviewer #1: Yes

Reviewer #2: Yes

Reviewer #3: Yes

3. Has the statistical analysis been performed appropriately and rigorously?

Reviewer #1: Yes

Reviewer #2: Yes

Reviewer #3: Yes

4. Have the authors made all data underlying the findings in their manuscript fully available (please refer to the Data Availability Statement at the start of the manuscript PDF file)?

Reviewer #1: No

Reviewer #2: Yes

Reviewer #3: No

5. Is the manuscript presented in an intelligible fashion and written in standard English?

Reviewer #1: Yes

Reviewer #2: Yes

Reviewer #3: Yes

Reviewer #1: Thank you for proactively addressing the reviewers' comments. No further comments on my end.

Reviewer #2: The authors have addressed all the review comments. The paper makes a good contribution, is strong and well-presented.

Reviewer #3: (No Response)

**Do you want your identity to be public for this peer review?** For information about this choice, including consent withdrawal, please see our Privacy Policy .

Reviewer #1: No

Reviewer #2: No

Reviewer #3: No
